# 1 Xact625i vs PX-375: A Comparative Study of Online XRF

# 2 Ambient Multi-Metal Monitors vs ICP-MS

- Laurence C. Windell<sup>1,3</sup>, Saliou Mbengue<sup>2</sup>, Petra Pokorná<sup>1</sup>, Jaroslav Schwarz<sup>1</sup>, André S.H.
- Prévôt³, Manousos I. Manousakas³,⁴, Stefanos Papagiannis⁴, Jakub Ondráček¹, Roman
- Prokeš<sup>2,5</sup>, Vladimir Ždímal<sup>1</sup>
- <sup>1</sup>Aerosol Chemistry and Physics Research Group, Institute of Chemical Process Fundamentals of the Czech
- Academy of Sciences, Prague, 16500, Czech Republic
- <sup>2</sup>Global Change Research Institute of the Czech Academy of Sciences, Brno, 60 300, Czech Republic
- <sup>3</sup>PSI Center for Energy and Environmental Sciences, Paul Scherrer Institut, Villigen PSI, 5232, Switzerland
- <sup>4</sup>Environmental Radioactivity & Aerosol Technology for Atmospheric & Climate Impact Lab, Institute of
- Nuclear and Radiological Science and Technology, Energy & Safety, NSCR Demokritos, Athens, 15310, Greece
- <sup>5</sup>RECETOX, Faculty of Science, Masaryk University, Brno, 61137, Czech Republic
- Correspondence to: Laurence C. Windell (laurence.windell@psi.ch)

33

39

#### Abstract

- 16 This study provides a comprehensive evaluation of the Xact625i and PX-375 online energy dispersive X-ray
- 17 fluorescence (ED-XRF) instruments for real-time trace elemental monitoring at a rural background site over six
- 18 months. This represents the first direct comparison between the two instruments, assessing their performance
- against inductively coupled plasma mass spectrometry (ICP-MS), the reference method for elemental analysis.
- 20 Both instruments demonstrated strong measurement capabilities, with the Xact625i achieving higher sensitivity
- for trace metals and showing closer agreement with ICP-MS using 24-hour averaging of 2-hour data ( $r^2 = 0.89$  vs
- 0.78 for the PX-375). The PX-375 reported higher overall concentrations, primarily due to overestimations of Si
- and S. Both instruments correlated well for Ca, Fe, Zn, and Pb, while detection limits affected Ni and Cd. The
- Xact625i exhibited superior performance in measuring elements such as S, V, and Mn. While correlations with
- ICP-MS were high, systematic over- and underestimations in absolute concentrations were found, particularly for
- the PX-375. When directly comparing the two online instruments using raw 2-hour data, a strong agreement was
- observed (mean  $r^2 = 0.95$ ). However, systematic slope discrepancies persisted, in line with comparisons against
- ICP-MS. The findings confirm the reliability of these two ED-XRF instruments for high-time-resolution elemental
- monitoring as a complementary technique to traditional filter analysis, enabling detailed source apportionment
- studies, improved trend analysis, and more responsive air quality management strategies. Future work comparing
- studies, improved trend analysis, and more responsive an quanty management strategies. Future work comparing
- ED-XRF to laboratory-based methods could refine harmonisation efforts and address systematic differences in
- absolute concentrations.

# Short Summary

- In this work, we compare the two most widely used online XRF monitors for ambient elemental analysis, the
- Xact625i and PX-375. We found strong correlations between the online instruments and the reference method
- (better so for the Xact625i), while in terms of absolute concentrations, some elements were over- and
- underestimated. Overall, we determined both instruments can be used as powerful tools to produce high-time
- resolution elemental data for use in air quality monitoring.

#### 1 Introduction

- Atmospheric particulate matter (PM) consists of organic and inorganic aerosol species such as sulphate, nitrate,
- ammonium, mineral compounds, and trace elements. Although low in mass contribution, trace elements (metals,
- metalloids, or non-metals) are a significant component of atmospheric particulate matter (PM). Unlike many
- organic and inorganic compounds, most trace elements are only slightly affected by atmospheric processes during
- their transport and can therefore be used as conservative tracers (Lim et al. 2010; Morawska and Zhang 2002) of
- natural (e.g., Al, K, Na, and Ti) and anthropogenic (e.g., Sb, Cu, and Ba for road traffic; As, Cu, Fe, Mn, and Pb

for steelworks and non-ferrous metallurgy; Cr, Se, As, and Co for coal combustion; Ni and V for fuel and oil combustion; and Cd, Pb, and Zn for waste incineration) sources. Ambient trace metals are linked to both chronic and acute effects on humans, causing the monitoring of these elements to be of great importance (Daellenbach et al. 2020; Fang et al. 2017; Shiraiwa et al. 2017). Some elements, such as As, Cd, Cr, and Ni, are considered carcinogenic to humans, while Pb, Sb, and Co are suspected of being probable carcinogens (WHO 2024). In the EU, annual limit values were established for Pb (0.5 ng m<sup>-3</sup>) and annual target values for Ni (20 ng m<sup>-3</sup>) and As (6 ng m<sup>-3</sup>), as maintained in the most recent Directive (EU 2024/2881).

Over the last decades, several monitoring and scientific research programmes, including ACTRIS (Aerosol, Clouds, and Trace Gases Research Infrastructure Network), have been implemented for the provision of high-quality harmonised data and information on various air pollutants, including trace metals. Ambient elemental analysis of PM has traditionally been carried out using the method of sample collection and subsequent filter analysis. Laboratory-based analyses, such as inductively coupled plasma-mass spectrometry (ICP-MS) and optical emission spectrometry (ICP-OES) and particle-induced x-ray emission (PIXE) boast high accuracy and reliability. ICP-MS and atomic absorption spectroscopy (AAS) are currently the only reference methods according to EU directives established for sampling ambient elements (EU Directive 2024/2881), with ICP-MS generally seen as the gold-standard (Ogrizek, Kroflič and Šala 2022), thus leading to the choice of this methodology for the current study.

However, filter collection often lacks high-time resolution, with filter sampling usually lasting 12 or 24 hours. Additionally, the process of filter collection, preparation, and analysis can not only be laborious and time-consuming but may also be the source of contamination or loss of material (Alleman et al. 2010), thus increasing the need for both time and resources. The introduction of online Energy Dispersive X-ray Fluorescence (ED-XRF), namely the Xact625i Ambient Multi-Metals Monitor (Cooper Environmental Services, OR, USA) and the PX-375 Continuous Particulate Monitor (Horiba, KY, Japan), has opened a new avenue for elemental analysis. ED-XRF offers a green analytical chemistry approach for multi-element analysis of airborne PM (Bilo, Cirelli and Borgese 2024).

The two instruments have been successfully deployed worldwide, from low-concentration background sites in Europe to high-pollution hotspots in India and China, and for a wide range of research applications, such as shortterm measurements of brake wear emissions to monitoring of domestic heating (Bilo, Cirelli and Borgese 2024; Faisal et al. 2025; Furger et al. 2017; Li et al. 2017; Lopez et al. 2023; Mach et al. 2022; Miyakawa et al. 2023; Rai et al. 2020; Tremper et al. 2018; Windell et al. 2024). These studies showed the instruments' capabilities to analyse pollution events (such as fireworks, wildfires, and dust events), capture key markers of emissions (such as Pb and As spikes near smelting facilities, V and Ni from shipping). The highly time-resolved data provided by these instruments can be combined with source apportionment to identify and chemically characterise key emission sources and their respective contributions to ambient metal concentrations and air quality trends. The results from monitoring efforts, spanning both short-term campaigns and long-term observational studies, are also crucial for the health sector, providing exposure data that can improve epidemiological studies, toxicological assessments, and public health interventions. The ability to provide hourly elemental exposure data enhances timeresolved health impact assessments, linking short-term spikes to hospital admissions and long-term trends to chronic diseases. By identifying and quantifying toxic elements, these findings help medical researchers and policymakers understand the long-term health impacts of airborne metal exposure. These results can inform emission reduction strategies in metallurgy, manufacturing, and transportation, leading to more targeted and effective air quality management policies. Recent developments have further expanded the applications of these instruments. For example, combining ED-XRF with oxidative potential measurements provides a direct link between elemental composition and toxicity, offering new insights into public health impacts (Camman et al. 2023; Cheung et al. 2024). Additionally, the introduction of switching inlets for size-resolved sampling allows for the differentiation of fine vs. coarse mode contributions from different sources at a single site (Furger et al. 2020; Manousakas et al. 2022).

This paper critically compares the two leading instruments for online elemental composition analysis of PM. As the primary options available, understanding their relative performance, strengths, and limitations is crucial for potential users. This study equips policymakers, researchers, and monitoring organisations with practical insights to guide instrument selection based on specific needs, such as monitoring capabilities, operational constraints, or data quality requirements. While the Xact625i and the PX-375 have been compared to reference techniques (Furger et al. 2017; Lopez et al. 2023; Trebs et al. 2024; Tremper et al. 2018), a direct comparison of these two

instruments has yet to be published and is highly needed within the scientific community. In addition, to our knowledge, this study offers the most comprehensive evaluation of the PX-375 against a reference method. The present study evaluates the Xact625i and PX-375's performances by comparing them with ICP-MS filter analysis, the current EU reference method (EN 14902). The two instruments were deployed side-by-side for six months in Central Europe. Furthermore, selecting a background site with low elemental concentrations allows for evaluating these instruments near their limits of detection.

#### 2 Methods

#### 2.1 Sample site and instruments

The measurements were conducted from 28/05/2023 to 30/11/2023 at the National Atmospheric Observatory Košetice (NAOK), a rural background site run by the Czech Hydrometeorological Institute (CHMI), and part of the EMEP (European Monitoring and Evaluation Programme), GAW (Global Atmosphere Watch), ICOS (Integrated Carbon Observation System), GMOS (Global Mercury Observation System), AERONET (Aerosol Robotic Network), and ACTRIS networks. It is located in the Czech highlands (49°34'24.13" N, 15°40'49.67" E, 534 m a.s.l.) surrounded by agricultural land and forests (Figure 1). The station is influenced by emissions from a road with low traffic activity located 1 km from the station, a biomass furnace 8 km away, and domestic heating in multiple nearby villages. A detailed description of the sampling site can be found in (Mbengue et al. 2023). Online measurements of trace elements were performed simultaneously at NAOK with the Xact625i and the PX-375, with an overview of instrument specifications found in Table S1. Both instruments were fitted with identical PM<sub>10</sub> heads (Tisch, USA) placed at 4 m a.g.l. and measurements were performed with flow rates of 16.7 (+/- 1%) lpm with two-hour time resolution. The air flows were heated to 45°C to keep the relative humidity below 45%. Two sets of collocated 24-hour offline filters were available and used for laboratory analysis using ICP-MS and ion chromatography (IC).

Figure 1. Location of the NAOK measurement site.

#### 2.2 Online ED-XRF instruments

143

#### 2.2.1 Xact625i ambient multi-metals monitor

- The Xact625i is an online ED-XRF spectrometer based on EPA IO method 3.3: determination of metals in ambient 130 particulate matter using X-ray fluorescence (XRF spectroscopy). Particles are deposited onto Teflon tape for 131 analysis. Sampling and analysis occur in parallel; that is, the tape advances from the sampling inlet to the XRF 132 block for simultaneous elemental analysis. The instrument was calibrated to measure 41 elements, namely Al, Si, 133 P, S, Cl, K, Ca, Sc, Ti, V, Cr, Mn, Fe, Co, Ni, Cu, Zn, Ga, Ge, As, Se, Br, Rb, Sr, Y, Mo, Ag, Cd, In, Sn, Sb, Te,
- Cs, Ba, La, Ce, Pt, Au, Hg, Tl, Pb, and Bi, although many of these are typically below limits of detection (LoD).
- The Xact uses an X-ray source (Rhodium anode, max 50kV and 50W) used to illuminate samples on the tape. 135
- Excited photons are collected by a silicon drift detector (SDD), with subsequent spectra analysed by a spectral
- analysis package. Three XRF excitation conditions (EC) were used with different voltages, currents, primary 138 filters, and analysis times to enable the analysis of a wide range of elements: EC1 (Al filter, 25 kV, 2000 μA, 20%
- analysis time), EC2 (Pd filter, 48 kV, 2000 µA, 40% analysis time), and EC3 (Cu filter, 48 kV, 2000 µA, 40%
- analysis time). For this study, EC3, which has the same parameters as EC2 aside from the type of primary filter,
- was only used to measure Cd. The instrument (serial number 626-00-200701G) was run using software versions
- of Xact Control 1.2.2.138, CES Spectrum v2.01, XRF TRF V1.34. 142

# 2.2.2 PX-375 continuous particulate monitor

- The PX-375 measures and records particulate mass and element concentration, combining the principles of beta 145 ray attenuation (for the mass analyser) and ED-XRF spectrometry (for the element analyser). The PX-375 mass 146 analyser continuously measures the mass concentration of particles deposited on a non-woven PTFE fabric filter
- tape during the sampling time with <sup>14</sup>C as a beta-ray source (maximum energy of 0.156 MeV), after which the 147
- sample is transferred to the element analyser, where the element concentration is measured. The X-ray tube is a 149 vacuum tube consisting of a tungsten filament, a target, and a beryllium window. The PX-375 was calibrated to
- measure 16 trace elements, namely Al, Si, S, K, Ca, Ti, V, Cr, Mn, Fe, Ni, Cu, Zn, As, Pb, and Cd. The instrument
- uses two XRF excitation conditions (EC): 15 kV for light elements (Al to Ca) without a primary filter, and 50 kV
- for the remaining elements using a Mo primary filter. The measurement conditions were set to factory settings
- and were the same as for calibration: a current of 200 µA and an analysis time of 500 s. The 500 s analysis time
- is traditionally chosen for the instrument to extend the lifetime of the X-ray tube, however, a longer time (e.g.,
- 1000 s) improves LoDs. The instrument was run using hardware revision number: MSIP-REM-HOR-144PX-375
- (revision date 26.09.2022) and software version PX-375 V1.3.2.37

#### 157 2.3 Calibration check, quality control

- XRF calibration checks were performed before, during, and after the measurement campaign for both instruments.
- The XRF calibration check involves the evaluation of the XRF system's performance by measuring against
- elemental standards. For the Xact625i, Micromatter calibration foils (Micromatter, Canada) were used (Table S2).
- The standards contain known amounts of elements, with one sample being used per element. For the PX-375, a
- multi-element calibration standard sample SRM2783 from NIST (National Institute of Standards & Technology,
- USA) was used, except for Cd, which was calibrated using a separate Micromatter calibration foil provided by
- the manufacturer (Table S3).
- The Xact625i conducts two automated quality assurance (QA) checks daily at midnight, lasting 15 minutes each.
- The first check, 'Energy Calibration', entails the use of a Cr and Nb-coated rod entering the XRF block, with
- subsequent spectral peaks being checked. The second test, 'QA Upscale' uses Cr, Pb, Cd, and Nb, with variations 167
- in levels of these elements above 10% producing an error message. Average standard errors relative to daily QA
- checks were Cr (1.02%), Nb (0.56%), Cd (0.39%), and Pb (0.80%) from a total of 181 QA checks.
- For the PX-375, X-ray intensity checks and energy calibrations were performed every month to maintain the
- performance of the instrument, using energy calibration standard samples provided by the manufacturer. The
- intensity checks were performed by the analysis of Cu intensity with voltage, current, and analysis times of 50
- 173 kV, 200 μA, and 500 s, respectively. Energy calibrations were performed by checking the (span point) spectrum
- for the Al standard. The Cu intensity and Al span point were measured within ±5% with respect to the standard
- values (-0.2% and 0.3%, respectively).

- Filter tape blank checks were performed before, during, and after the measurement campaign (Table S4). The
- checks entailed the analysis of a blank HEPA filter sample (sample inlet connected to a HEPA filter). For the Xact,
- three XRF excitation conditions were used, set to default analysis times of 1440 seconds for EC1 and 2880 seconds
- for EC2 and EC3. A total of six tests (two at 240 min sampling time and four at 120 min) were carried out. For
- the PX-375, six blanks were also measured with 120 min sampling, and the analyses were performed in the same
- condition as for the calibration (500 s, 200  $\mu$ A, and 15 kV/50 kV).
- Instrument LoDs are found in Table S5. Harmonisation of both instruments' LoD calculations was not possible;
- in point of fact, repeat field blank concentrations were, for most elements, too low to calculate LoDs. For the
- Xact625i, LoD values were taken as double the manufacturer's interference-free 1σ detection limits (68%
- Confidence Level (C1σ) (EPA, 1999), i.e., determined element by element; interference-free LoDs are inherently
- underestimated. For the PX-375, LoDs were taken from the manufacturer's 2σ given LoDs for S, Ti, Cr, Mn, Pb,
- and Cd, and where not possible, calculated from blank measurements, e.g., for Si, K, and Ca, or taken as averages
- from three pieces of literature, e.g., for Al, V, Fe, Ni, Cu, Zn, and As (Creamean et al. 2016; Miyakawa et al. 2023;
- Yang et al. 2023). Since methods for LoD calculation are different for both instruments, a comparison of the
- instruments' sensitivities is not possible.

# 191 2.4 Error propagation

- The Xact software provides dynamic uncertainties for each data point, specific to individual elements,
- incorporating spectral interferences, volume flow uncertainty, and summation uncertainty.
- A flat 5% uncertainty is recommended by the manufacturer for the PX-375. Uncertainties were calculated
- **195** following Eq. (1):

$$U = \sqrt{(x \cdot a)^2 + (x \cdot b)^2 + (x \cdot 0.1)^2}$$
 (1)

- where x is concentration, a is average flow rate variation over the period, b is relative XRF error based on
- elemental standard analysis, and the 5% recommended flat uncertainty is doubled. The uncertainty evaluation for
- the PX-375 does not consider spectral interferences, unlike the Xact625i, and is thus potentially underestimated.
- For the offline ICP-MS analysis, uncertainties were calculated following Eq. (2): (2)
- $U = \sqrt{(x \cdot c)^2 + (x \cdot d)^2}$

212

- where x is concentration, c is uncertainty from repeated (5-time) measurements, and d is flat instrumental
- uncertainty specific to each element.

# 204 2.5 Investigation of raw spectra

- Raw spectra samples and blank tapes were investigated in order to identify whether potential differences between
- the two instruments' results could be attributed to deconvolution software or to the hardware components. The
- goal here was to observe differences in peak-to-background ratios between the two instruments by overlapping
- chosen data spectra with blank tape spectra. Spectra were evaluated with the PyMCA code (Solé et al. 2007) and
- the Origin Pro (OriginLab, MA, USA) software package to check for line overlaps and overall fitting performance.
- The specific fitting algorithms used by each instrument, however, are not disclosed, which could contribute to any
- observed discrepancies, and therefore the spectral analysis was solely used as an indicator.

# 2.6 Offline sampling and laboratory analysis

- A Leckel sampler (LVS-auto 2.3, Sven Leckel, Germany) equipped with a PM<sub>1</sub> inlet, operating at 2.3 m<sup>3</sup>/h for 24
- hours with 47 mm filters and automatic filter change, was used for filter collection. 19 mm punches were cut from
- 215 filters (quartz) and extracted in 5 mL of ultrapure water, shaken for 2 hours using a Kavalier LT 2 shaker (Kavalier,
- Czech Republic) with horizontal rectilinear movement (40 mm amplitude, 150 rpm). Filter vials were placed
- horizontally in parallel with the direction of movement, ensuring effective fibre disaggregation. The filters were
- conditioned according to ČSN EN 12341 gravimetric procedure and included blanks, with a moving average of
- 20 blanks used for subtraction. The extracts were passed through a Rotilabo<sup>®</sup> syringe filter (PVDF, unsterile, 0.22
  μm pore size, 30 mm diameter; ROTH, Germany) and analysed using a Dionex ICS-5000 system (Dionex, Ca,
- USA) both for cations (Na<sup>+</sup>, NH<sub>4</sub><sup>+</sup>, K<sup>+</sup>, Ca<sup>2+</sup>, and Mg<sup>2+</sup>) and anions (SO<sub>4</sub><sup>2-</sup>, NO<sub>3</sub><sup>-</sup>, and Cl<sup>-</sup>) in parallel, with injections
- of 25 μl to each column using a Dionex AS-AP autosampler. A Dionex IonPac AS11-HC (2x250 mm) column (a

precolumn Dionex IonPac AG11-HC Guard, 2x50 mm was preceded) was used for anions using potassium hydroxide as an eluent, produced by a Dionex cation EGC III KOH eluent generator (flow rate 0.380 mL/min) and a Dionex IonPac CS18 (2x250 mm) column (a precolumn Dionex IonPac CG18 Guard, 2x50 mm was preceded) for cations using a methane sulphonic acid solution as an eluent (flow rate 0.250 mL/min). Both anion and cation setups were equipped with electrochemical suppressors. External calibration was done using NIST traceable calibration solutions.

Separate filter collection for ICP-MS was carried out using a Leckel sampler (LVS-auto 2.3, Sven Leckel, Germany) equipped with a PM<sub>10</sub> inlet with a flow rate of 2.3 m<sup>3</sup>/h for 24 hours every second day and 47 mm nitrocellulose membrane filters (MF-Millipore<sup>TM</sup>, 3.0 μm pore size, hydrophilic), with blank correction based on a moving average of 20 blanks as described above. In addition, four field blanks were processed together with ICP-MS samples and used for blank subtraction during laboratory analysis. The ICP-MS analysis was carried out by the Transport Research Centre (CDV, Czech Republic), a research institution under the jurisdiction of the Czech Ministry of Transport. Filters were digested using an ETHOS One microwave digester (Milestone, Italy) with PTFE vessels in a mixture of 6 mL HNO<sub>3</sub> + 2 mL H<sub>2</sub>O<sub>2</sub> at a maximum temperature and power of 232 °C and 1000 W for 10 minutes. After digestion, the mixture was evaporated to near dryness and diluted to 20 mL with 3% HNO<sub>3</sub>. Analysis was run five times for repeatability, with four field blanks being used for blank subtraction. Elements quantified were Al, Si, S, K, Ca, Ti, V, Cr, Mn, Fe, Ni, Cu, Zn, As, Cd, Pb, Ba, and Sn using an Agilent 8800 ICP-MS/MS operated in collision/reaction cell mode. Most elements were measured on-mass in He mode (27Al, 39K, 43Ca, 47Ti, 51V, 52Cr, 55Mn, 56Fe, 60Ni, 63Cu, 66Zn, 111Cd, 118Sn, 137Ba, 208Pb); Si was acquired in O2 reaction mode (on-mass), and S and As were measured by mass-shift as SO<sup>+</sup> (32  $\rightarrow$  48) and AsO<sup>+</sup> (75  $\rightarrow$  91), respectively. An internal-standard mix (Bi, Ge, In, 6Li, Sc, Tb, Y; 10 µg L<sup>-1</sup> in 3% HNO<sub>3</sub>) was added to all standards and samples. Daily tuning used a working solution (Ce, Co, Li, Mg, Tl, Y) prepared to 1 µg L<sup>-1</sup>. Calibration employed three multi-point curves: (i) Ti, V, Cr, Ni, Cu, Zn, As, Cd, Pb, Ba, Sn at 0–100 µg L<sup>-1</sup>; (ii) K, Ca at 0-10 mg L<sup>-1</sup>; and (iii) Al, Si, S, Mn, Fe at 0-10 mg L<sup>-1</sup>. Trueness was verified with CRMs: A mixture of 23 elements (Analytika, CZ), SLRS-6, ERM-TM-23.5, and ERM-ION-96.4. Operating parameters were RF power 1550 W, He flow 4.0 mL min<sup>-1</sup>, O<sub>2</sub> flow 0.3 mL min<sup>-1</sup> (low-matrix method). Typical recovery rates for metals were 80-110%. The 2-hour online data from both instruments were averaged against 24-hour offline ICP-MS filter data covering the period between 28/05/2023 and 30/11/2023 (N = 45).

# 252 3 Results and Discussion

#### 3.1 Accuracy and precision of Xact625i and PX-375

The SRM2783 was tested with the PX-375 (Table S3), and the relative accuracy to reference material was obtained in the range of 75% for Cu to 112% for Mn. The Micromatter standard foils were analysed by the Xact625i with a relative error within ± 2% (recovery rate > 98%) for all 16 elements (Table S2). Higher accuracy in results found when measuring reference materials by the Xact625i over the PX-375 can be attributed to the high concentrations of the Micromatter foils used for calibration of the former instrument; the analysis of these single-element standards entails a lack of spectral inferences, compared to the PX-375's SRM2783, a combined multi-standard with lower concentrations than those found in the Micromatter foils, and does not necessarily indicate low precision of the PX-375. It is notable, however, that relative errors for the PX-375 were generally skewed in the negative direction (average -5.4% relative error). However, the deviation observed when analysing the SRM2783 by the PX-375 opens a discussion as to the calibration method of the instrument.

The Micromatter elemental standards were also analysed using the PX-375 to provide insights into the PX-375's performance. Relative accuracy (recovery) was calculated as  $100 \times (\text{measured / reference})$ ; for foils, this was  $100 \times (\text{PX-375})$  measured value on the Xact625i Micromatter foil / nominal foil value). Due to the large size of the SRM2783 foil, it could not be analysed by the Xact625i. Results of the Micromatter calibration foils tested with the PX-375 can be seen in Fig. 2. On average, elements were underestimated by 17%, with relative accuracy dropping to sub ~60% for Cr, As, and Cu. Two large outliers, V and S, had relative accuracies of 14.5% and 139.6%, respectively. Similarly, in Trebs et al. (2024), multi-element reference materials from the UC Davis Air Quality Research Centre (UCD-47-MTL-ME-233 and UCD-47-MTL-ME-234) have been tested using a PX-375 initially calibrated with the SRM2783, and significant differences were observed between measured and reference values for Ti (>100%), V (23-24%), As (30-37%), Al (9-64%), Si (35-52%), and S (57-60%) (Trebs et al. 2024).

An additional goal was to determine if relative accuracy when measuring reference materials reflected field measurements in comparison to reference ICP-MS (Fig. 2). For field data, relative accuracy was calculated as 100 × (PX-375 24-hour daily average / collocated ICP-MS 24-hour daily average). While relative accuracy for some elements reflected differences against ICP-MS (e.g., S, K, Ti, Mn, Fe, Zn, and Pb), significant differences were not overall observed. In addition, as these calibration standards were originally used for calibration of the Xact625i, and not the PX-375 (although the PX-375 shares an identical individual standard for Cd, alluding to the viability of the use of these standards in both instruments), these results may only be used as an indication.

Figure 2. Recovery rates from the analysis of the Xact625i Micromatter standard foils using the PX-375 and field measurements against ICP-MS.

#### 3.2 Data treatment/limits of detection

The highest LoDs were obtained for the light elements such as Al, Si, and S for Xact625i and Al, K, and Ca for the PX-375 (Table S5). Xact625i shows lower LoDs (except for Al and Si), and this could be explained by the shorter recommended analysis time used for the PX-375 (500 s) compared to Xact625i (1440 – 2880 s).

The final results of the data cleaning process are shown in Table S6. For some elements (e.g., Cd and Ni for both instruments, and As and Cu for the PX-375), the majority of samples (>90%) were below online instrumental LoDs and for these elements, comparisons against offline data may not be robust. Elements such as V and Cr that had 60-80% of data points below LoDs were kept in the analysis, but their low data coverage was taken into account when attempting to draw conclusions from their results. In addition, field measurements uncovered high-concentration peaks across crustal elements for the PX-375, but were not detected by the Xact625i or reflected in the offline analysis despite the close proximity of the instruments. Samples with over 5 times greater concentrations in chosen elements (Ca and Zn) were flagged. By inspecting images recorded by the metal-oxide-semiconductor (CMOS) camera of the PX-375, the presence of insects and other coarse debris (in the range of a few mm) was detected, leading to contamination for most of the flagged samples (Figure S1). Thus, a total of 101 samples of 1888 (5.34%) were removed from the dataset, with two of the 45 daily averaged samples for comparison with ICP-MS also being removed. Here, the presence of the built-in camera proves useful, especially for quality assuring samples with suspicious data.

# 3.3 Beta attenuation validation (PX-375)

 $PM_{10}$  mass concentrations are monitored by the CHMI at NAOK using an MP101M particulate analyser (Environnement S.A., Paris, France) based on beta attenuation (gravimetric-equivalent), running at one-hour time resolution, thus directly comparable to the PX-375's in-built beta attenuation analysis. For PX-375, the flagged samples due to contamination (as described in the previous section) were three times larger than the reference (over 40  $\mu$ g/m³) and were omitted for a better comparison.

The PX-375's beta attenuation measurements were compared to the station's reference instrument 2-hour average  $PM_{10}$  (time series in Figure S2), with an  $R^2$  correlation of 0.42 (Figure S3), and total average concentrations of  $10.53\pm7$  and  $11.05\pm7.6$  µg/m³, respectively. Averaging over 24 h greatly increased the correlation factor ( $R^2=0.82$ , Figure S4), with daily averages of  $9.8\pm5.2$  and  $10.7\pm5.7$  µg/m³ for the PX-375 and the reference instrument, respectively.

# 3.4 Online measurements against offline ICP-MS analysis

 $PM_{10}$  measured by the station's reference method showed values over the 43 samples ranging from 4.5 to 22.8  $\mu g/m^3$ . For the 43 daily averaged samples, total elemental concentrations for the PX-375 and Xact625i were 1772 and 1182  $ng/m^3$  respectively, compared to 1245  $ng/m^3$  for the offline analysis, skewed by overestimation of more abundant elements (e.g., Al and S) by the PX-375. An overview of elemental concentrations is shown in Fig. 3 and in Table S7.

Figure 3. Cumulative elemental concentrations in percentage (left axis) and total concentration (right axis) for each methodology for the six-month time period.

Results were obtained from Deming regressions, which take the analytical uncertainty of both variables into account (Deming 1943), with intercepts fixed at zero as the main results, and can be seen in Table 1 with median concentrations shown in Table S7. Time series can be found in Fig. S5. Excluding Ni, Cu, and Cd, Pearson r<sup>2</sup> values for the PX-375 and Xact625i ranged from 0.63 to 0.97 (mean of 0.78) and 0.69 to 0.97 (mean of 0.89),

respectively. To investigate potential offsets, a separate analysis using Deming regressions fitted with intercepts was run, with a summary table found in Table S8 and plots in Fig. S6. Since both ED-XRF instruments are physically expected to have a zero–zero relationship with ICP-MS (i.e., no concentration should correspond to no signal), regressions forced through the origin were retained as the main comparison approach, with two cases (Pb and As) investigated separately.

Table 1. Deming regression results and Pearson correlation coefficients for ED-XRF against offline ICP-MS (24-hour averages) and direct intercomparison of ED-XRF vs ED-XRF (2-hour data). Uncertainties ( $\pm$ ) represent one standard error from 2000 bootstrap resamples; 95% confidence intervals  $\approx$  estimate  $\pm$  1.96  $\times$  SE.

|            | PX-375 vs ICP-MS (n = 43) |                | Xact625i vs ICP-MS (n = 43) |                | Xact625i vs PX-375 (2-hour data) |                |           |
|------------|---------------------------|----------------|-----------------------------|----------------|----------------------------------|----------------|-----------|
| Element    | Slope                     | r <sup>2</sup> | Slope                       | r <sup>2</sup> | Slope                            | r <sup>2</sup> | N samples |
| Al         | 2.19 ±<br>0.14            | 0.86           | 0.38 ±<br>0.10              | 0.8            | 1.09 ±<br>0.19                   | 0.96           | 189       |
| Si         | 1.89 ±<br>0.17            | 0.84           | 1.31 ±<br>0.08              | 0.89           | 1.26 ± 0.16                      | 0.96           | 1111      |
| S          | 1.57 ±<br>0.02            | 0.97           | 0.93 ±<br>0.01              | 0.97           | 0.59 ± 0.01                      | 0.99           | 1800      |
| K          | 0.58 ±<br>0.03            | 0.65           | 0.93 ±<br>0.04              | 0.83           | 1.64 ± 0.13                      | 0.97           | 1722      |
| Ca         | 0.76 ±<br>0.04            | 0.85           | 0.86 ±<br>0.02              | 0.96           | 1.15 ±<br>0.08                   | 0.9            | 1775      |
| Ti         | 0.86 ±<br>0.21            | 0.71           | 2.22 ±<br>0.11              | 0.91           | 1.59 ±<br>0.13                   | 0.98           | 888       |
| V          | 0.35 ±<br>0.06            | 0.71           | 0.63 ±<br>0.08              | 0.88           | 1.34 ± 0.16                      | 0.98           | 238       |
| Cr         | 1.42 ±<br>0.16            | 0.63           | 0.56 ±<br>0.05              | 0.69           | 0.65 ± 0.1                       | 0.94           | 423       |
| Mn         | 0.52 ±<br>0.03            | 0.65           | 0.69 ±<br>0.03              | 0.88           | 1.63 ± 0.12                      | 0.98           | 1025      |
| Fe         | 0.70 ±<br>0.04            | 0.77           | 0.90 ±<br>0.02              | 0.94           | 1.37 ± 0.15                      | 0.98           | 1751      |
| Zn         | 0.89 ±<br>0.03            | 0.81           | 0.85 ±<br>0.02              | 0.96           | 0.99 ±<br>0.05                   | 0.88           | 1747      |
| Pb         | 0.55 ±<br>0.06            | 0.88           | 0.42 ±<br>0.05              | 0.95           | 0.65 ± 0.03                      | 0.83           | 722       |
| As*        |                           |                | 1.85 ±<br>0.12              | 0.95           |                                  |                | 31        |
| Cu**       |                           |                | 0.53 ±<br>0.06              | 0.07           |                                  |                | 197       |
| Ni*        |                           |                |                             |                |                                  |                | 18        |
| Cd*        |                           |                |                             |                |                                  |                | 17        |
| Average*** | 1.03 ±<br>0.09            | 0.78           | 0.96 ±<br>0.07              | 0.89           | 1.16 ±<br>0.11                   | 0.95           |           |

<sup>\*</sup>Omitted due to an excessive number of samples below LoD. \*\*A number of outliers skew Cu results for the Xact625i; with 10 outliers removed, a slope of 0.5 and an  $r^2$  value of 0.76 remains. \*\*\*Excluding Cu for the Xact625i.

The lightest elements measured by ED-XRF, especially Al and Si, showed significantly different slopes compared to ICP-MS, with Si being overestimated by both instruments, and over- and underestimation in terms of Al for the PX-375 and Xact625i, respectively (Figure 4). However, these elements for both instruments were highly correlated with ICP-MS, potentially alluding to systematic differences in absolute concentration levels. Moving up in atomic number, elements S (an important element considering atmospheric sulphate content) and Ca (often used as soil resuspension markers) showed good results against the reference method. However, PX-375

overestimated S (slope of 1.57), while K showed both underestimation and a lower correlation (R<sup>2</sup> of 0.65) and thus did not perform as well. For Ti (commonly used to detect dust), the Xact625i showed a better correlation with ICP-MS (R<sup>2</sup> of 0.91 vs 0.71) but exhibited significant overestimation (slope of 2.18).

Analysing light elements with ED-XRF can be challenging due to several factors. In ambient samples containing a range of elements, higher uncertainties are generally expected for elements with potential spectral interferences, and those more prone to self-absorption effects, especially the lightest elements such as Si, S, K, and Ca (Furger et al. 2017). During our campaign, S, associated with particulate sulphate, was the most abundant element in the samples, accounting for 40.3%, 35.3% and 36.1% of the total elemental mass measured by PX-375, Xact625i and ICP-MS, respectively. When S concentration is high, elements such as Al and Si may be hard to quantify accurately with ED-XRF due to spectral interferences from the large nearby, more intense S peak and matrix effect, which may lead to over- or underestimation in Al and/or Si (Indresand and Dillner 2012). Therefore, the spectral deconvolution, calculations of the peak area, and corrections for peak overlap and matrix effects are key factors for accurate determination in the elemental mass concentrations.

Figure 4. Deming regression charts forced through origin for ED-XRF instruments against ICP-MS for lighter elements.

In terms of toxic trace elements, Cr proved difficult to quantify by the ED-XRF instruments, owing to low concentrations at NAOK often near instrumental LoDs, and consequently high uncertainties, with similar results for V for the PX-375 yet better  $r^2$  and regression slope for the Xact625i (Figure 5). Pb was well measured by both instruments in terms of high  $r^2$  values, while underestimating absolute concentrations by a factor of  $\sim$ 0.5, with indications of nonlinearity to be examined further below.

Figure 5. Deming regression charts forced through origin for ED-XRF instruments against ICP-MS for toxic trace elements.

Both instruments proved well in measuring wear-associated elements, with Fe and Zn with regression slopes closer to unity (Figure 6), and showing high  $r^2$  values (with the Xact625i edging the PX-375 in both metrics). While the Xact625i showed a strong correlation for Mn,  $r^2$  for the PX-375 was reduced, and underestimation of the transition element was observed for both instruments.

Figure 6. Deming regression charts forced through origin for ED-XRF instruments against ICP-MS for wear-associated elements.

Two elements, As and Cu, were most of the time undetected by the PX-375 due to concentrations being below instrumental LoD. The shorter analysis time for the PX-375 (500 s) may explain the inability of the instrument to measure these elements. Line interference is well-known for the couple Pb–As, and the higher concentration of Pb observed in the samples during the campaign can also make detection of As difficult for PX-375 (Furger et al., 2017). The Xact625i measured As with a high r² value of 0.95 but coupled with high overestimation (Figure 7), with indications of nonlinearity to be examined further below. Ten outliers were observed for Cu, skewing r² to 0.07 from a potential 0.76 when removed; due to the absence of PX-375 Cu data, the source disagreement for these data points cannot be determined.

Figure 7. Deming regression charts forced through origin for the Xact625i against ICP-MS for elements only measurable by the Xact625i.

Results from the separate non-forced fit function showed that for most elements, regression slopes were not significantly different to those determined by forcing through 0.0. Gradient slope for Cr by the PX-375 improved from 1.42 to 1.03, and V slopes for the PX-375 and Xact625i improved from 0.35 to 0.54 and 0.63 to 0.84, respectively, however due to high uncertainties and high spread of the data for these elements, these gradient slopes are not representative of good agreement. For Pb, gradient slopes improved from 0.55 to 0.91 for the PX-375 and 0.42 to 0.73 for the Xact625i, and for As, slope improved from 1.85 and 1.39 for the Xact625i (Figure 8). The inclusion of intercepts for these two elements may provide meaningful insight. Both instruments showed negative intercepts for Pb (-0.26 for PX-375 and -0.37 for Xact625i), suggesting a slight underestimation at very low ICP-MS concentrations. In contrast, As showed a positive intercept (+0.23 for Xact625i), consistent with a small baseline offset in the XRF measurement. These behaviours can partly be explained by the large relative uncertainty of very low-concentration points, which tend to drive the regression away from the origin and generate apparent offsets. For Pb and As, however, the fitted intercepts also capture a systematic feature of the XRF response across the dataset, rather than a simple statistical artifact. Incorporating intercepts therefore improves representativeness of the regression for these elements, whereas for most others the forced-through-origin approach remains more appropriate.

Figure 8. Deming regression charts with intercept for ED-XRF against ICP-MS for two elements displaying non-linear fits.

A study on the earlier Xact625 model at an urban site against ICP-MS showed comparable results in terms of  $r^2$  values to this study (average  $r^2$  values for elements available for comparison of 0.95 vs 0.93, respectively) (Tremper et al. 2018). While regression slopes for Fe and Zn were similar, and better agreement in this study found for As (although still overestimated), Tremper et al. (2018) found regression slopes for elements V, Cr, Mn, and Pb of 0.87, 0.99, 1.10, and 1.02, compared to 0.63, 0.56, 0.69, and 0.42 in our study likely owing to higher concentrations found in their urban study. However, similar agreements in terms of  $r^2$  indicate the ability of the instrument to work similarly well at rural sites, such as the current sample site, as with urban sites.

Another study on the Xact625 at a rural traffic-influenced site compared to ICP-MS showed comparable r² values to elements also detected in the current study, aside from V and As (Furger et al. 2017); here, lower r² values compared to this study were found for these two elements (0.57 and 0.50 vs 0.88 and 0.95, respectively), as well as their study finding a V slope gradient of 0.15 against our value of 0.63 and an As slope gradient of 0.56 against 1.85, further highlighting the difficulty of measuring these trace elements. Ti seemed to be similarly overestimated in both studies (gradient slope of 2.22 in our study and a slope of 1.13 combined with a significant intercept of +5.58 in the referred study). Considering high intercept values in linear regression equations in Furger et al. (2017), elements Mn, V, Cr, and Pb were significantly overestimated in their study compared to underestimation in the present study. These results indicate instrumental uncertainty for elements in low concentrations, considering both studies were conducted at rural sites.

A study on the PX-375 at a traffic site compared Fe, Ti, and Zn against reference XRF methodology (Lopez et al. 2023). Zn was comparable for both studies, while significantly better results were found in the present study for Fe (slopes of 0.50 vs 0.70, r<sup>2</sup> values of 0.60 vs 0.77) and for Ti (slopes of 0.65 vs 0.86, r<sup>2</sup> values of 0.44 vs 0.71).

These notable differences, especially considering lower elemental concentrations in the present study at a rural site, may be explained by the lower sample size measured by Lopez et al. (2023).

The comparison of ED-XRF results with ICP-MS highlighted differences in performance between the two online instruments, particularly regarding over- and under-estimations. Elements with a difference greater than 40% compared to ICP-MS were considered significantly different, with this threshold used to categorise the elements as either over- or under-estimated. The PX-375 significantly overestimated four elements (Al, Si, S, and Cr) and underestimated four others (K, V, Mn, and Pb), while the Xact625i overestimated two elements (Ti and As) and underestimated three others (Al, Cr, and Pb); here, it must be considered that elements such as Al, Cr, and Pb were close to limits of detection. However, r² values may be considered a more reliable performance metric, as different methodologies (such as the comparison to ICP-MS here) may produce varying absolute concentrations as well as differing on an element-by-element basis. The PX-375 had an acceptable average r² value of 0.78, while the Xact625i demonstrated superior performance with an average r² value of 0.89. Generally good agreement was found within the literature for the Xact625i, indicating measurement consistency, aside from elements found in concentrations closer to LoDs. Although better results for the PX-375 were found in this study against the previous study used for comparison, further studies analysing the instrument are required for robust comparisons.

# 3.5 Online ED-XRF against offline IC analysis

Sulphate and potassium ions measured by IC were compared against their online counterparts, with regressions shown in Fig. 9 and time series in Figs. S7 and S8.

Figure 9. Deming regression charts forced through origin for online ED-XRF instruments against reference ion chromatography analysis.

Elemental S was compared to water-soluble PM<sub>1</sub> SO<sub>4</sub><sup>2-</sup> (24-hour averages, n = 85) after division of the respective ion by three. SO<sub>4</sub><sup>2-</sup> was strongly correlated with both instruments' elemental S ( $r^2$  values of 0.94 and 0.95 for the Xact625i and PX-375, respectively), with overestimation reflected by both ED-XRF instruments.

Potassium, while mostly in forms insoluble in water when sourced from dust and soil resuspension (Andreae 1983), is soluble when emitted from biomass burning (Yu et al. 2018). While  $K^+$  from the major ion analysis was typically not correlated with its elemental counterpart, it was so during the heating season (n = 20).  $K^+$  was well correlated with elemental K for the Xact625i ( $r^2$  of 0.97) and less so for the PX-375 ( $r^2$  of 0.59), with overestimation again reflected by both ED-XRF instruments.

While the regression slopes from the IC analysis indicated similar findings to the comparison of ED-XRF and ICP-MS, i.e., higher S concentrations found by the PX-375 and lower for the Xact625i and vice versa for K, inconsistencies may be due to organic S not measured by IC, sources of insoluble K, and the offline sampling being done in PM<sub>1</sub> compared to PM<sub>10</sub> (although soluble S and K are known to be mostly found <1  $\mu$ m in diameter (Huang et al. 2016; Zhang et al. 2012)).

#### 3.6 Direct PX-375 vs Xact625i analysis

 Two-hour data for the full dataset, treated the same as 24-hour averaged data against ICP-MS (i.e., removing datapoints <LoD and those affected by contamination, see Section 3.2), were directly compared between the two instruments. Here, the goal was to compare the instruments at high-time resolution (as opposed to averaging to 24 hours for comparison to ICP-MS filter data), and to confirm whether ED-XRF vs ED-XRF results agree with the ED-XRF vs ICP-MS analysis, with results in Table 1. For this comparison, the 2-hour resolution was retained rather than averaging to 24 hours, as this maximises sample size (up to 1800 data points), avoids uncertainty from incomplete 24-hour coverage, and provides a more representative test of the instruments at their native operating resolution, with results for both analyses seen in Table S9.

Results from the ED-XRF vs ED-XRF showed higher r² values compared to the lower average values found from PX-375 vs ICP-MS (0.78 vs 0.95), and similar to that of Xact625i vs ICP-MS (0.89). A likely reason is due to the ED-XRF analysis being based on a much larger dataset (up to 1800 samples) of 2-hour resolution samples, compared to ED-XRF vs ICP-MS being based on 43 daily-averaged samples, thus reducing statistical noise. Secondly, while ICP-MS has been proven here as a strong reference method due to high r² values, the consistent measurement method of ED-XRF is likely to lead to better agreement when compared directly. Higher r² values in the PX-375 vs Xact625i comparison were found compared to ED-XRF vs ED-XRF across all elements. In particular, r² values increased significantly for elements K, Ti, V, Cr, Mn, and Fe for the PX-375, and K and Cr for the Xact625i, with those already high in correlations remaining similar. Low sample numbers must be noted for Al, V, and Cr, and results for these elements must be treated with caution. When comparing regression slopes for elements with sufficient sample number after treatment, similar trends for ED-XRF vs ED-XRF were shown compared to ED-XRF vs ICP-MS, where overestimation by, e.g., the Xact625i against ICP-MS was indicated by a lower slope gradient in the Xact625i vs PX-375 analysis, with a paired t-test showing no significant difference.

The fact that the analysis of the full 2-hour dataset showed slopes remaining consistent across comparisons (e.g., overestimated S for the PX-375 vs ICP-MS also true for PX-375 vs Xact625i), confirms that bias due to inconsistent methodologies is minimal, reinforcing confidence in the conclusions drawn from the ED-XRF vs ICP-MS analysis. Here, as was the case for ED-XRF vs ICP-MS, regression slopes far from unity again indicate systematic differences in absolute concentrations between the two instruments. The increase in R² values for PX-375 when compared against Xact625i (rather than ICP-MS) can be attributed to the higher sample size and time resolution in the ED-XRF vs ED-XRF comparison, where daily averaging smooths out short-term variations, thus weakening correlations. The strong agreement between the two instruments further reinforces their ability to track elemental trends consistently, highlighting that despite over- and underestimations, they capture temporal variability in a comparable manner.

# 3.7 Comparison and analysis of XRF spectra

The comparative analysis revealed notable differences in the elemental concentrations reported by the two instruments. Table 2 describes elemental concentrations for a datapoint chosen for investigation which filled the criteria of having mostly non-zero values across elements and large differences for multiple elements of concern identified from the ED-XRF vs ICP-MS comparison, such as overestimation of lighter elements by the PX-375.

Table 2. Elemental concentrations for the chosen 08-06-2023 06:00 datapoint.

| Element | PX-375 (ng/m <sup>3</sup> ) | Xact625i (ng/m³) |
|---------|-----------------------------|------------------|
| Al      | 361.5                       | 0                |
| Si      | 488.3                       | 271              |
| S       | 1521.6                      | 788.4            |
| K       | 136.6                       | 145.5            |
| Ca      | 127.9                       | 123.8            |
| Ti      | 1.1                         | 9.4              |
| V       | 0.1                         | 0.1              |
| Cr      | 0.4                         | 0.4              |
| Mn      | 3.4                         | 3.4              |
| Fe      | 156.9                       | 136.3            |
| Ni      | 1.2                         | 0                |
| Cu      | 6.2                         | 1.8              |
| Zn      | 13.7                        | 11               |
| As      | 0                           | 1.7              |
| Cd      | 0                           | 0                |
| Pb      | 4.1                         | 1.7              |

Figs. 10a and b illustrate the differences in spectra obtained from the two instruments under the energy conditions (EC) used for analysing light elements (15kV, no filter for the PX-375, and EC1, Al filter for the Xact625i). For elements S to Ca, no differences in peak-to-background ratios are visible despite discrepancies in concentrations of Al, Si, and S, potentially indicating the differences in concentrations are software-based for these elements.

Figure 10. (a) Above: PX-375 XRF spectrum for the sample 08-06-2023 06:00 and the corresponding blank for the energy condition 15 kV, and (b) below: Xact625i XRF spectrum for the same datapoint and the corresponding blank for EC1 (25 kV).

Figure 11 presents the XRF spectra for ECs set to analyse mid- to heavier elements (50 kV, Mo filter for the PX-375, and EC2 Pd filter for the Xact625i). Figure 11a (PX-375) reveals that the blank spectrum exhibits significant spectral artefacts, originating from the instrument itself rather than the blank Teflon tape. In contrast, the Xact625i delivers spectra with improved clarity in the energy range of 6.0 to 14.0 keV, exhibiting minimal noise for the aforementioned elements (Figure 11b). These differences are reflected in the measured concentrations; we observe significantly different concentrations for Ti, Ni, As, and Pb between the two instruments. Here, concentration discrepancies between the instruments may be attributed to hardware-related issues.

Figure 11. (a) Above: Horiba PX-375 XRF spectrum for the sample 08-06-2023 06:00 and the corresponding blank for the energy condition 50 kV, Mo filter, and (b) below: Xact625i XRF spectrum for the same datapoint and the corresponding blank for EC2 (48 kV, Pd Filter).

#### **4 Conclusion**

This study, based on a long-term (6-month) campaign at a rural background site, is the first of its kind, providing a comprehensive evaluation of the Xact625i and PX-375 instruments for online measurements of ambient trace elements. Overall, both instruments showed strong performance for key elements, though the Xact625i exhibited higher sensitivity for trace metals and better agreement with the reference method. Results were particularly strong considering sampling at a rural site with generally low elemental concentrations, demonstrating the instruments' abilities to measure at lower concentrations for many elements.

When comparing with the reference methodology (ICP-MS), the Xact625i slightly outperformed the PX-375, with stronger correlations (r² values of 0.89 vs 0.78) and greater variation in elements against ICP-MS. r² values for the PX-375 ranged from 0.63 to 0.97, while those for the Xact625i ranged from 0.69 to 0.96. Total average concentration for the PX-375 was higher than that of the Xact625i (1772 and 1182 ng/m³, compared to 1245 ng/m³ for ICP-MS) skewed by higher readings for more abundant elements (e.g., Si and S having gradient slopes against the reference of 1.89 and 1.57, compared to 1.31 and 0.93 for the Xact625i). Both ED-XRF instruments performed well in measuring elements such as Ca, Fe, and Zn, with gradient slopes ranging from 0.7 to 0.9 and r² values ranging from 0.77 to 0.96. The Xact625i seemed to achieve higher sensitivity due to better measurements of trace elements found in low concentrations at the site (e.g., V with an Xact625i gradient slope of 0.63 and r² value of 0.88, PX-375 slope of 0.35 and r² value of 0.71 and Mn with an Xact625i gradient slope of 0.69 and r² value of 0.88, PX-375 slope of 0.52 and r² value of 0.75). Performance of both instruments fell off for elements Ni and Cd, which were undetectable due to LoDs, with the PX-375 additionally being unable to measure As. Fitted intercepts revealed systematic offsets for Pb (-0.26 and -0.37 for the PX-375 and Xact625i, respectively) and As (0.23 for

- the Xact625i), suggesting a tendency towards underestimation for Pb and a baseline offset for As, likely reflecting 556 a combination of baseline shifts in the XRF response and high uncertainty at low concentrations.
- Direct comparisons of ED-XRF vs ED-XRF using 2-hour data showed strong internal consistency (R<sup>2</sup> = 0.95),
- higher than PX-375 vs ICP-MS (0.78) and Xact625i vs ICP-MS (0.89) indicating that lower correlations in ICP-
- MS comparisons were at least partly due to daily averaging rather than measurement inconsistencies. However,
- regression slopes largely aligned with ICP-MS trends, validating elemental over- and underestimation patterns for
- both instruments.
- While many elements were strongly correlated with ICP-MS, significant differences in absolute concentrations,
- such as for S, K, and Mn (PX-375) and Pb and Ti (Xact625i), indicate systematic differences in concentration
- levels for these measurements. Similar over- and underestimations greater than 50% compared to reference
- measurements were also found for the PX-375 and Xact625i in previous studies (Furger et al. 2017; Lopez et al.
- 2023), while better results were found in a traffic-heavy site compared to the present study (Tremper et al. 2018),
- indicating stronger performance of the Xact625i in heavier polluted sites. Correction factors based on these
- differences may be investigated where r<sup>2</sup> values remained high. An increased analysis time for the PX-375 (e.g., 569 from 500 s to 1000 s) would improve LoDs, especially important when analysing trace elements such as Cd and
- As, at the expense of a shorter X-ray tube lifetime. A strong recommendation is to complement online ED-XRF
- with filter-based analysis at each specific sample site for data quality control to compare systematic differences
- in absolute concentrations.
- Variations observed between the instruments could be attributed to differences in the X-ray tubes and energy
- conditions used by each spectrometer, the analysis time used for the PX-375, and the distinct fitting algorithms
- employed for both instruments, such as regions of interest, fundamental parameters, or peak deconvolution.
- Results for the PX-375 could also be attributed to the multi-element standard approach to calibration, as well as
- by results from the spectral analysis, revealing suboptimal background-to-noise ratios for elements measured
- above 6 keV.
- One of the key advantages of online ED-XRF instruments is their ability to provide high-time-resolution elemental
- data, which is crucial for pollution event detection, source apportionment, and exposure assessments. The findings
- from this study prove the abilities of the PX-375 and Xact625i to produce robust, continuous online data, given
- the high importance of real-time trace metal monitoring, particularly in regions where rapid emission changes can
- impact air quality and public health. Further work must be done comparing ED-XRF to reference methodologies
- such as ICP-MS and laboratory-based XRF, as well as larger-scale intercomparisons between online instruments,
- and to further optimise operation and calibrations.

Data availability. The datasets generated and analysed in this study will be deposited in Zenodo and made publicly available upon publication of the article at https://doi.org/10.5281/zenodo.17198675

590 591

587

Author contributions. LCW: Conceptualisation, Methodology, Validation, Formal Analysis, Investigation, Writing - Original Draft, Visualisation. SM: Conceptualisation, Methodology, Validation, Formal Analysis,

Investigation, Writing – Original Draft, Visualisation. PP: Conceptualisation, Supervision, Validation, Writing – Review & Editing. JS: Formal Analysis, Validation, Writing - Review & Editing. ASHP: Supervision, Funding

Acquisition, Validation, Writing – Review & Editing. MIM: Validation, Writing – Review & Editing. SP:

Formal Analysis, Writing - Review & Editing. JO: Writing - Review & Editing. RP: Funding Acquisition,

Writing – Review & Editing. VZ: Supervision, Funding Acquisition, Validation, Writing – Review & Editing. 598

Disclaimer. This manuscript has not been published and is not under consideration for publication elsewhere.

601

Acknowledgement. This work was supported by the Ministry of Education, Youth and Sports of the Czech Republic under grant of large research infrastructure ACTRIS-CZ (LM2023030), by CzechGlobe investment for support of research infrastructures (108916), and by the MI-TRAP project as part of the EU Horizon European

programme under grant agreement No. 101138449. P. Pokorná, J. Schwarz, J. Ondráček, and V. Ždímal also 604 605

thank the project of Czech Science Foundation No. 24-10768S for financial support.

Competing Interests. The authors declare that they have no conflict of interest.

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
