# Peer review of "Xact625i vs PX-375: A Comparative Study of Online XRF"

_EGUsphere, 2025_

## Author Comment (AC1)

[revised manuscript text omitted]

**Figure 3̶4. Deming regression charts ̲f̲o̲r̲c̲e̲d̲ ̲t̲h̲r̲o̲u̲g̲h̲ ̲o̲r̲i̲g̲i̲n̲ for ED-XRF instruments against ICP-MS for lighter elements.**

In terms of toxic trace elements, Cr proved difficult to quantify by the ED-XRF instruments, owing to low concentrations at NAOK often near instrumental LoDs, and consequently high uncertainties, with similar results for V for the PX-375 yet better $r^2$ and regression slope for the Xact625i (Figure 4̶5). Pb was well measured by both instruments in terms of high $r^2$ values, while underestimating absolute concentrations by a factor of ~0.5̲,̲ ̲w̲i̲t̲h̲ ̲i̲n̲d̲i̲c̲a̲t̲i̲o̲n̲s̲ ̲o̲f̲ ̲n̲o̲n̲l̲i̲n̲e̲a̲r̲i̲t̲y̲ ̲t̲o̲ ̲b̲e̲ ̲e̲x̲a̲m̲i̲n̲e̲d̲ ̲f̲u̲r̲t̲h̲e̲r̲ ̲b̲e̲l̲o̲w̲.

[revised manuscript text omitted]

Flow rate checks

Flow rate checks on the Xact625i and PX-375 were performed using a Gilibrator 2 air flow calibrator (Sensidyne, USA) once per month. When the flow rates varied by over 1% compared to the reference flow meter, a flow calibration was carried out. Average variation in flow rates over the campaign for both instruments was <1%.

Table S1. Overview of technical specifications for the two ED-XRF instruments.

| Spec | Xact 625i (Cooper Environmental) | PX-375 (Horiba) |
|---|---|---|
| X-ray source | Rh anode tube, max 50 kV / 50 W | W tube (vacuum, Be window), 15 / 50 kV auto switch |
| Primary filters | 3 (Al, Pd, Cu) | 2 (none, Mo) |
| Detector | Silicon Drift Detector (SDD, ED-XRF) | Silicon Drift Detector (SDD, ED-XRF) |
| Sampling | Teflon filter tape, 16.7 L/min | PTFE filter tape, 16.7 L/min, $\beta$-ray attenuation + XRF, CMOS spot imaging |
| Elements measured | Up to 40+ (Al–Bi, configurable) | 16 standard (Al–Cd), 25+ with extended calibration |
| Time resolution | 15 min – 4 h selectable, analysis time identical to sampling time | 0.5 – 24 h selectable (500–10000 s analysis time selectable) |
| Calibration / QA | Calibration with SRMs; Internal reference checked each sample; daily auto-drift | Factory calibration with SRM |
| Power | ~2 kW, 120/220 VAC | ~400 W, 100–240 VAC |
| Size / Weight | ~48×51×76 cm / ~83 kg | ~43×55×29 cm / ~40 kg |
| Distinguishing | Wide element coverage, EPA-validated | Dual PM mass + metals, compact, low power, CMOS dust-spot imaging |

Table S2. XRF calibration check using the Xact625i single-element calibration standards. The XRF tests were carried out before, during, and after the campaign.

| Elements | Analysis time (s) | Concentration (ng/m³) | | | Recovery rate | Relative Error |
|---|---|---|---|---|---|---|
| | | Reference | measured (n = 3) | | | |
| | | | mean | std | | |
| Al | 4400 | 62143 | 61677 | 130.83 | 99.25% | -0.75% |
| Si | 2700 | 50375 | 49728 | 133.45 | 98.72% | -1.28% |
| S | 1200 | 17665 | 17337 | 91.80 | 98.14% | -1.86% |
| K | 300 | 28098 | 27930 | 53.57 | 99.40% | -0.60% |
| Ca | 120 | 30785 | 30414 | 182.13 | 98.79% | -1.21% |
| Ti | 60 | 65171 | 64851 | 225.66 | 99.51% | -0.49% |
| V | 60 | 61356 | 61015 | 100.11 | 99.44% | -0.56% |
| Cr | 60 | 69430 | 69020 | 93.73 | 99.41% | -0.59% |
| Mn | 60 | 62822 | 62104 | 412.49 | 98.86% | -1.14% |
| Fe | 60 | 56768 | 56510 | 55.29 | 99.55% | -0.45% |
| Ni | 70 | 53038 | 52418 | 135.37 | 98.83% | -1.17% |
| Cu | 70 | 56612 | 56337 | 61.03 | 99.51% | -0.49% |
| Zn | 120 | 19544 | 19400 | 9.49 | 99.26% | -0.74% |
| As | 70 | 38473 | 37803 | 82.18 | 98.26% | -1.74% |
| Cd | 2700 | 36632 | 36505 | 10.70 | 99.65% | -0.35% |
| Pb | 70 | 62029 | 61385 | 216.13 | 98.96% | -1.04% |

Table S3. XRF calibration check using the SRM2783 multi-element calibration standard (*cadmium single-element standard tested separately). The XRF tests were carried out before, during, and after the campaign.

| Elements | Concentration (ng/m³, *ug/cm²) | | | Recovery rate | Relative Error |
| --- | --- | --- | --- | --- | --- |
| | Reference | Analysis time = 500 s, measured (n = 3) | | | |
| | | mean | std | | |
| Al | 2064.52 | 2142.18 | 134.16 | 104% | 4% |
| Si | 5395.52 | 5405.92 | 245.59 | 100% | 0% |
| S | 113.51 | 99.56 | 13.85 | 88% | -12% |
| K | 490.33 | 495.41 | 10.60 | 101% | 1% |
| Ca | 1288.58 | 1282.07 | 17.93 | 99% | -1% |
| Ti | 144.71 | 138.78 | 3.25 | 96% | -4% |
| V | 5.43 | 5.21 | 1.29 | 96% | -4% |
| Cr | 17.31 | 15.14 | 3.89 | 87% | -13% |
| Mn | 27.55 | 30.97 | 1.42 | 112% | 12% |
| Fe | 2548.63 | 2530.52 | 4.51 | 99% | -1% |
| Ni | 6.00 | 5.13 | 1.30 | 85% | -15% |
| Cu | 43.57 | 32.85 | 6.54 | 75% | -25% |
| Zn | 173.36 | 168.62 | 4.79 | 97% | -3% |
| As | 1.10 | 0.84 | 0.31 | 76% | -24% |
| Cd* | 11.20 | 10.87 | 0.15 | 97% | -3% |
| Pb | 35.21 | 35.02 | 4.44 | 99% | -1% |

Table S4. Repeat blank tape measurements for both instruments with concentrations in ng/m$^3$. Analysis times were 120 min for the Xact625i and 500 s for the PX-375 to harmonise with field measurements.

| Element | Xact625i (n = 6) | | | PX-375 (n = 10) | | |
|---|---|---|---|---|---|---|
| | Mean | Mean unc | std | Mean | Mean unc | std |
| Al | 24.48 | 60.54 | 75.14 | | | |
| Si | | | | 1.15 | 0.05 | 1.44 |
| S | 0.54 | 1.42 | 2.23 | | | |
| K | | | | 3.46 | 0.17 | 3.82 |
| Ca | | | | 9.95 | 0.49 | 5.43 |
| Ti | 0.030 | 0.162 | 0.064 | 0.356 | 0.018 | 0.858 |
| V | 0.010 | 0.063 | 0.041 | | | |
| Cr | | | | 0.303 | 0.015 | 0.232 |
| Mn | | | | 0.019 | 0.001 | 0.053 |
| Fe | | | | | | |
| Ni | | | | 0.010 | 0.001 | 0.028 |
| Cu | | | | | | |
| Zn | | | | | | |
| As | 0.001 | 0.006 | 0.006 | | | |
| Cd | 0.174 | 0.573 | 1.166 | 1.885 | 0.094 | 2.339 |
| Pb | | | | 0.018 | 0.001 | 0.042 |

Table S5. Instrumental Limit of Detections (LoDs) in ng/m$^3$. For the PX-375: * 3σ averages from literature values, ** 3σ field blank measurements, and *** 2σ manufacturer values.

| Element | Xact625i (1σ) interference-free LoD (doubled from manufacturer values) | PX-375 (mixed) |
|---|---|---|
| Al | 122.57 | 40.10* |
| Si | 21.72 | 4.32** |
| S | 3.85 | 6.60*** |
| K | 1.43 | 11.46** |
| Ca | 0.36 | 16.29** |
| Ti | 0.19 | 5.00*** |
| V | 0.15 | 0.42* |
| Cr | 0.14 | 0.50*** |
| Mn | 0.17 | 2.20*** |
| Fe | 0.21 | 2.49* |
| Ni | 0.12 | 0.57* |
| Cu | 0.1 | 0.93* |
| Zn | 0.08 | 1.06* |
| As | 0.08 | 2.01* |
| Pb | 0.16 | 2.40*** |
| Cd | 3.08 | 10.40*** |

Table S6. Final data coverage after data treatment procedure. Starred elements omitted from analysis due to low coverage.

| Element | Sample coverage (n samples) | |
| --- | --- | --- |
| | PX-375 | Xact625i |
| Al | 1294 (68.6%) | 189 (10.0%) |
| Si | 1787 (94.6%) | 1111 (58.9%) |
| S | 1800 (95.3%) | 1800 (95.3%) |
| K | 1722 (91.2%) | 1793 (94.9%) |
| Mn | 1775 (94.0%) | 1783 (94.4%) |
| Fe | 888 (47.0%) | 1780 (94.3%) |
| Ca | 238 (12.6%) | 405 (21.5%) |
| Ti | 423 (22.4%) | 595 (31.5%) |
| V | 1025 (54.3%) | 1515 (80.2%) |
| Cr | 1751 (92.7%) | 1787 (94.7%) |
| Zn | 1764 (93.4%) | 1747 (92.5%) |
| Pb | 722 (38.3%) | 890 (47.1%) |
| As* | 3 (0.2%) | 96 (5.1%) |
| Cu* | 40 (2.1%) | 1526 (80.9%) |
| Ni* | 43 (2.3%) | 18 (1.0%) |
| Cd* | 18 (1.0%) | 2 (0.1%) |

Table S7. Descriptive data from campaign measurements of the 43 daily averaged samples with concentrations in ng/m$^3$.

| Element | Median concentration across field measurements (+/- uncertainty) | | |
| | PX-375 | Xact625i | ICP-MS |
| --- | --- | --- | --- |
| Al | 272.08 (+/-) 40.11 | 63.81 (+/-) 416.21 | 134.83 (+/-) 21.98 |
| Si | 480.01 (+/-) 61.61 | 343.76 (+/-) 692.47 | 261.18 (+/-) 44.36 |
| S | 715.92 (+/-) 119.22 | 417.41 (+/-) 7.76 | 448.49 (+/-) 16.42 |
| K | 64.71 (+/-) 7.76 | 87.87 (+/-) 3.46 | 94.80 (+/-) 13.31 |
| Ca | 107.42 (+/-) 13.67 | 121.21 (+/-) 4.87 | 142.26 (+/-) 30.63 |
| Ti | 8.54 (+/-) 1.48 | 11.73 (+/-) 1.14 | 5.62 (+/-) 0.78 |
| V | 0.20 (+/-) 0.05 | 0.19 (+/-) 1.25 | 0.30 (+/-) 0.05 |
| Cr | 0.62 (+/-) 0.14 | 0.23 (+/-) 1.91 | 0.34 (+/-) 0.05 |
| Mn | 2.52 (+/-) 0.55 | 2.63 (+/-) 0.74 | 3.90 (+/-) 0.44 |
| Fe | 111.33 (+/-) 14.58 | 125.68 (+/-) 4.71 | 143.05 (+/-) 22.73 |
| Zn | 5.87 (+/-) 0.68 | 5.51 (+/-) 0.31 | 6.33 (+/-) 1.32 |
| Pb | 1.22 (+/-) 0.18 | 0.63 (+/-) 0.82 | 1.37 (+/-) 0.14 |
| As* | | 0.79 (+/-) 0.17 | 0.40 (+/-) 0.06 |
| Cu* | | 0.87 (+/-) 0.21 | 1.85 (+/-) 0.38 |
| Total | 1772 | 1182 | 1245 |

*Starred samples omitted due to too many samples <LoD for the PX-375 for As and Cu.*

Table S8. Deming regression results and Pearson correlation coefficients for ED-XRF vs ICP-MS i) forced through origin and ii) with intercept. Uncertainties (±) represent one standard error from 2000 bootstrap resamples; 95% confidence intervals ≈ estimate ± 1.96 × SE.

| Element | ED-XRF vs ICP-MS (Deming, forced through origin) | | PX-375 vs ICP-MS (Deming with intercept) | | Xact625i vs ICP-MS (Deming with intercept) | | N samples |
|---|---|---|---|---|---|---|---|
| | PX375 slope (0,0) | Xact625i slope (0,0) | Slope | Intercept | Slope | Intercept | |
| Al | 2.19 ± 0.14 | 0.38 ± 0.10 | 2.2 ± 0.46 | -0.95 ± 26.73 | 0.26 ± 0.56 | -2.55 ± 53.84 | 189 |
| Si | 1.89 ± 0.17 | 1.31 ± 0.08 | 1.89 ± 0.39 | 0.34 ± 34.95 | 1.53 ± 0.17 | -45.52 ± 13.15 | 1111 |
| S | 1.57 ± 0.02 | 0.93 ± 0.01 | 1.67 ± 0.08 | -33.22 ± 23.81 | 0.98 ± 0.04 | -15.54 ± 13.05 | 1800 |
| K | 0.58 ± 0.03 | 0.93 ± 0.04 | 0.81 ± 0.15 | -15.61 ± 9.35 | 1.17 ± 0.23 | -18.75 ± 15.58 | 1722 |
| Ca | 0.76 ± 0.04 | 0.86 ± 0.02 | 0.69 ± 0.1 | 5.83 ± 6.21 | 0.93 ± 0.08 | -5.4 ± 7.31 | 1775 |
| Ti | 0.86 ± 0.21 | 2.22 ± 0.11 | 0.93 ± 0.51 | -0.13 ± 0.51 | 2.18 ± 0.32 | 0.13 ± 0.43 | 888 |
| V | 0.35 ± 0.06 | 0.63 ± 0.08 | 0.54 ± 0.12 | -0.02 ± 0.01 | 0.84 ± 0.2 | -0.08 ± 0.04 | 238 |
| Cr | 1.42 ± 0.16 | 0.56 ± 0.05 | 1.03 ± 0.25 | 0.12 ± 0.08 | 0.56 ± 0.1 | -0.0 ± 0.02 | 423 |
| Mn | 0.52 ± 0.03 | 0.69 ± 0.03 | 0.6 ± 0.09 | -0.19 ± 0.21 | 0.82 ± 0.12 | -0.52 ± 0.27 | 1025 |
| Fe | 0.70 ± 0.04 | 0.90 ± 0.02 | 0.75 ± 0.11 | -2.89 ± 3.97 | 0.93 ± 0.06 | -2.07 ± 2.23 | 1751 |
| Zn | 0.89 ± 0.03 | 0.85 ± 0.02 | 1.0 ± 0.12 | -0.5 ± 0.52 | 1.03 ± 0.07 | -0.91 ± 0.41 | 1747 |
| Pb | 0.55 ± 0.06 | 0.42 ± 0.05 | 0.91 ± 0.19 | -0.26 ± 0.15 | 0.73 ± 0.09 | -0.37 ± 0.08 | 722 |
| As* | | 1.85 ± 0.12 | | | 1.39 ± 0.14 | 0.23 ± 0.05 | 31 |
| Cu** | | 0.53 ± 0.06 | | | 0.53 ± 0.16 | 0 ± 0.21 | 197 |
| Ni* | | | | | | | 18 |
| Cd* | | | | | | | 17 |
| Average*** | 1.02 ± 0.11 | 0.89 ± 0.12 | 1.03 ± 0.24 | | 0.95 ± 0.13 | | |

*Omitted due to an excessive number of samples below LoD. **A number of outliers skew Cu results for the Xact625i; with 10 outliers removed, a slope of 0.5 and an $r^2$ value of 0.76 remains. ***Excluding Cu for the Xact625i.

Table S9. Deming regression results and Pearson correlation coefficients for ED-XRF vs ED-XRF using i) 2-hour data and ii) 24-hour averages. Uncertainties (±) represent one standard error from 2000 bootstrap resamples; 95% confidence intervals ≈ estimate ± 1.96 × SE.

| Element | Xact625i vs PX-375 (2-hour data) | | | Xact625i vs PX-375 (24-hour averages) | | |
|---|---|---|---|---|---|---|
| | Slope | $r^2$ | N samples | Slope | $r^2$ | N samples |
| Al | 1.09 ± 0.19 | 0.96 | 189 | 1.42 ± 0.4 | 0.47 | 58 |
| Si | 1.26 ± 0.16 | 0.96 | 1111 | 0.93 ± 0.12 | 0.84 | 143 |
| S | 0.59 ± 0.01 | 0.99 | 1800 | 0.60 ± 0.01 | 0.99 | 174 |
| K | 1.64 ± 0.13 | 0.97 | 1722 | 1.32 ± 0.11 | 0.89 | 174 |
| Ca | 1.15 ± 0.08 | 0.9 | 1775 | 0.86 ± 0.15 | 0.70 | 173 |
| Ti | 1.59 ± 0.13 | 0.98 | 888 | 1.05 ± 0.27 | 0.69 | 146 |
| V | 1.34 ± 0.16 | 0.98 | 238 | 0.66 ± 0.22 | 0.45 | 72 |
| Cr | 0.65 ± 0.1 | 0.94 | 423 | 0.34 ± 0.05 | 0.24 | 99 |
| Mn | 1.63 ± 0.12 | 0.98 | 1025 | 1.06 ± 0.26 | 0.35 | 165 |
| Fe | 1.37 ± 0.15 | 0.98 | 1751 | 1.14 ± 0.17 | 0.79 | 174 |
| Zn | 0.99 ± 0.05 | 0.88 | 1747 | 0.80 ± 0.12 | 0.90 | 174 |
| Pb | 0.65 ± 0.03 | 0.83 | 722 | 0.98 ± 0.06 | 0.74 | 134 |
| Average | 1.16 ± 0.11 | 0.95 | | 0.93 ± 0.16 | 0.67 | |

[Figure]

Figure S1. Insect and other debris in the sampling spot of the PX-375 in samples flagged as outliers due to high concentrations of crustal elements.

[Figure]

Figure S2. PM$_{10}$ two-hour time series for reference and PX-375 in-built gravimetric analyses.

[Figure]

Figure S3. PM$_{10}$ two-hour linear regression scatter plot for reference and PX-375 in-built gravimetric analyses.

[Figure]

Figure S4. PM$_{10}$ daily-averaged linear regression scatter plot for reference and PX-375 in-built gravimetric analyses.

[Figure]

[Figure]

[Figure]

[Figure]

[Figure]

[Figure]

[Figure]

Figure S5. Elemental time series for the Xact625i and PX-375 against reference ICP-MS.

[Figure]

[Figure]

[Figure]

Figure S6. Deming regression charts for ED-XRF instruments against ICP-MS without forcing through origin.

[Figure]

Figure S7. Time series of elemental sulphur by ED-XRF instruments and offline corrected sulphate by ion chromatography.

[Figure]

Figure S8. Time series of elemental potassium by ED-XRF instruments and offline potassium cation by ion chromatography.

---

## Author Response (AR1)

**RC1:**

We would like to thank Reviewer 1 for suggestions that helped to improve the current study. We agreed with all the suggestions and as such, made the necessary changes/additions. Please find our responses to comments in blue.

**General Comments:**

"The analysis of the data seems quite superficial, in that while Deming regression is an appropriate test, the intercept is fixed at zero and nonlinearities are largely ignored. This to me is an omission, because many elements of interest such as Pb, Cr and As very clearly demonstrate nonlinear relationships in the scatter plots. Given the remit of the paper, these must be discussed both in terms of the possible causes and implications. It may also be prudent to perform and additional numerical fit to quantify this effect, such as using a different fit function or testing a nonzero intercept from the Deming regressions. A cursory inspection would indicate to me that if the intercept on arsenic in particular were to be allowed to vary, a much more favourable slope would be yielded (for whatever reason), so these effects cannot be ignored."

We agree that not including a secondary fit was an oversight. Therefore, a full re-analysis using Deming regressions without forcing through 0,0 was added to the supplement (Table S8 and Figure S6) and for two elements that indeed showed non-linear relationships, Pb and As, the intercept-fitted analyses were used in the main body as opposed to results forced through origin (Figure 8). Please see line 328 onwards in the track-changes document for the justification of retaining the origin-forced regressions as the main body (except for As and Pb), and line 388 onwards for the full analysis of non-origin forced regressions. Charts for the intercept-fitted regressions and a comparison of intercept vs non-intercept results are now found in the supplement.

"There needs to be more information regarding the offline sampling and analysis. While some of this may be considered obvious to those familiar with the reference methods, these must still be stated for the sake of those who are not. Specifically:

- More information on the Leckel samplers should be given (e.g. inlets, blank strategy)
- The manufacturer(s) and product codes of the filters used for offline analysis should be given, along with specification (in particular pore size) and any pre-treatment (e.g. washing, baking).
- The 'shaker' used for the aqueous extraction needs describing better and the product information on the syringe filter included.
- More information given on the extraction process used for ICP, in particular the use of a microwave digester (I think this is implied, but not stated) and the fact that the substrate is included in the digestion.
- While the digestion protocol is standard, the authors should still comment on the acid mixture used in the digestion and the likely recovery rate (particularly of silicates), noting that other mixtures based on aqua regia and hydrofluoric acid are used elsewhere."

Based on the suggestions, we have added thorough detail to every stage of the methods process as requested (section 2.6).

"The conclusions does not feature nearly enough quantitative information, with the comparison using descriptive terms instead. Given the slopes are crucially important, these should be referred to

and summarised quantitatively with reference to a range that the authors would consider acceptable. Furthermore, quantitative comparisons of these compared to previous equivalent literature should also be included."

We have added quantitative information to the conclusion, including explicit references to regression slopes and references to the previous literature (with the full comparison to previous literature found towards the end of section 3.6), strengthening this section.

**Specific comments:**

"Title: That the instruments are compared to ICP-MS as well is important because this represents an independent point of reference, so this should be incorporated into the title."

We have added 'vs ICP-MS' to highlight this comparison.

"Line 51: The authors should refer to the most recent directive (2024/2881), even though the limits were maintained from previous.

**Updated to current directive.**

Line 107: A map should be included, showing the location of the sampling site and context such as the distance from major pollution sources.

**Location of the measurement site added as suggested by the reviewer (Figure 1)**

Sections 2.2.1/2.2.2: Because the manufacturers regularly modify instruments, more information on the specific physical units used should be given here (ideally hardware revision numbers). Furthermore, because some of the discrepancies found are later attributed to software, the version numbers of the inversion software should be given.

Hardware revision numbers, software versions (lines 139-140 and 153-154) and technical specifications (Table S1) are added for each instrument.

Figure 1: The means by which this data was calculated should be more explicit. 'Relative Accuracy' is not very specific.

**Added in line 275.**

Line 272: 'Gravimetric' is not an appropriate term because this refers to the weighing of filters. Beta attenuation is a 'gravimetric equivalent' measurement.

**Corrected to gravimetric equivalent.**

Figure 2 (and elsewhere): "Total Concentration" should not be used because it could easily be confused with PM. "Total Detectable Concentration" would be more appropriate.

Axis title has been changed to Total Detectable Concentration.

Line 293: Worth stating that Deming regression takes the uncertainty of both variables into account. It should also be stated that the intercepts were fixed at 0 (may seem obvious, but it is important to state)."

**This has been clarified in lines 324 and 325 as well as under charts.**

"Figure 3 (and other equivalent plots): The use of grey makes the plots hard to read, in particular the whiskers. Suggest using different colours. It would also be a useful guide to the eye to include the 1:1 line on these plots as well."

Charts now use red instead of grey as well as having a 1:1 line for clarity.

"Line 329: "Both departments" is a very strange turn of phrase. Revise."

We have changed 'departments' to 'metrics'.

"Line 468: The term "offset" I would take to mean a systematic additive error, but the analysis performed here would be insensitive to this because the intercepts were fixed at zero. The authors should be more specific about what they mean here. If they are saying there is such an error, then the fits should be performed again without the constraint."

The reviewer is correct in saying the use of the term 'offset' is wrong here. Instead, it has been changed to 'systematic differences' which better suits the phenomenon seen here. The text now includes reference to the intercept-fitted analysis to better explain these observations.

"Line 488: Given that the measurement site was part of various networks including EMEP and ACTRIS, surely some of the data is in the public domain?"

As the Xact625i is not a 'standard' instrument (as opposed to filter-based elemental analyses whose data are public) for these networks, data are not published. However, upon discussion, we have decided to publicly upload the data, and as such, have changed the data availability statement: "Data availability. The datasets generated and analysed in this study will be deposited in Zenodo and made publicly available upon publication of the article at <a href="https://doi.org/10.5281/zenodo.17198675">https://doi.org/10.5281/zenodo.17198675</a>".

**RC2:**

We would like to thank RC2 for their comments and recommended suggestions. We have made the necessary changes, strengthening the paper for publication. Please find our answers in blue.

**Major comments:**

"Methods, Section 2.3: The Xact 625i also has an internal standard every hour – does a similar check exist for the PX-375? Please include the internal standard in a paragraph"

We confirm that the PX-375 in fact does not include an hourly standard, unlike the Xact625i.

"Methods/General: It might be useful to have a table with the main parameters compared for the two instruments to have an easy overview — can be in supplementary."

We agree with this suggestion and have added a comparative table of instrument specifications to the supplement (Table S1).

"Methods, Section 2.6: Sentence: More details are needed, such as filter manufacturer, blank levels, recovery etc. or paper reference to method if it is described with those details elsewhere or possibly reference method that is followed."

We have added through details for this methods section including all suggested additions.

"Results and Discussion, Section 3.1, L237 onwards: Is the analysis area the same in the Xact 625i and the PX-375? If not, does that need to be considered when using the Xact standards?"

We thank the reviewer for raising this point. The analysis areas of the two instruments are indeed different: the Xact625i analyses a smaller fixed spot on the Teflon tape, while the PX-375 scans a larger portion of the filter tape. This difference could contribute to minor variations when using standards. However, since the Micromatter foils used here are uniform across their surface, the different analysis areas do not affect the validity of the test.

"Results and Discussion, Table 1 and associated figures: Please include a confidence interval for the slope. Also, it might be useful to not have the intercept fixed at zero to see if the offset (with confidence interval) is significant."

We have added confidence to all slopes found in the paper. A full secondary analysis using Deming regressions with fitted intercepts has been added to investigate potential differences in the analyses. While the original analysis has been kept as the main analysis, the fitted intercept analysis was used for Pb and As that show nonlinear relationships. The fitted intercept charts and a table comparing results are found in the supplement.

"Results and Discussion, Section 3.6: I think this section would benefit if the comparison included 24hr means between the two ED-XRF methods, so it is more directly comparable to the filter comparison as well as the hight time resolution comparison."

We have added a table comparing results of the 2-hour and 24-hour averaged data in the Supplement. While we acknowledge that 24-hour averages provide direct comparability with offline filters, we chose to retain the 2-hour resolution as the main analysis, as this maximises sample size, avoids uncertainty from incomplete 24-hour coverage, and better represents the instruments' normal operating resolution (see line 457 in the tracked changes document).

"Introduction, L41: Sentence "Although low in weight contribution..." should be "Although low in mass contribution..."

**Corrected 'weight' to 'mass'.**

Introduction, L43: I would change this to "..., most elements are only slightly affected by..."."

**Corrected as suggested.**

"Methods – please review method subchapter numbers as there seems to be a jump from 2.1 to 2.2.1"

**A title "2.2 Online ED-XRF instruments" has been inserted.**

"Methods, L115: Sentence "Online trace elements were performed..." should be "Online measurements of trace elements were performed...""

**Thank you for pointing out this error, it has been amended.**

"Results and Discussion, L263: Sentence "...data points below the LoD were kept in the analysis must be treated..." should be "...data points below the LoD were kept in the analysis but must be treated...""

We have changed this sentence for clarity: "Elements such as V and Cr that had 60-80% of data points below LoDs were kept in the analysis, but their low data coverage was taken into account when attempting to draw conclusions from their results." (Line 292 onwards).

Here, the term 'treated' wrongly implied treatment of data – instead, these elements were treated the same as the rest of the data, but kept in the analysis despite low data coverage and as such were not used to make strong conclusions based on their results.

"Results and Discussion, L404 onwards: Sentence "Higer r2 values in the PX-375 vs Xact625i comparison were found compared to ED-XRF vs ED-XRF across all elements" please revise as I think it should be ED-XRF vs ICP-MS."

We understand that this sentence was not clear — we have clarified that this section is regarding the direct internal comparison of ED-XRF vs ED-XRF using 2-hour data.

"Results and Discussion, Figure 8: It might be worth showing the same energy range on the X-axis for easier comparison."

Here, the difference in energy range originates from the excitation conditions of the two instruments. The Xact625i EC1 operates at 25 kV, allowing excitation and detection of fluorescence lines up to  $^{\circ}6$  keV (e.g. Ti K $\alpha$  at 4.51 keV). In contrast, the PX-375 'EC1' operates at 15 kV, so effective excitation is limited to lighter elements, with measurable peaks only up to  $^{\circ}4.5$  keV. Beyond this energy range, the PX-375 spectra contain no signal of interest.